# VIDEO DIFFUSION MODEL FOR POINT TRACKING

## ABSTRACT

Point tracking aims to estimate pixel trajectories across video frames but remains challenging under large displacements, occlusion, and real-world artifacts. Conventional trackers, built on image-centric backbones and synthetic training, often fail in these settings. We revisit this problem through the lens of video diffusion models based on Diffusion Transformers (DiTs), whose 3D global attention structure and large-scale training naturally provide global temporal context and real-world priors. We first analyze the intrinsic robustness of video DiT features, showing stronger correlation maps than supervised ResNet backbones even under occlusion and motion blur. To fully exploit these properties, we introduce an upsampler that restores spatial detail while fusing multi-layer features, followed by an iterative refiner for high-precision trajectories. Extensive experiments on TAP-Vid benchmarks demonstrate that our framework achieves superior robustness and accuracy compared to existing backbones, establishing video DiTs as powerful foundations for point tracking.

## 1 INTRODUCTION

Point tracking (Doersch et al., 2023; 2022; Cho et al., 2024b; Karaev et al., 2024a) aims to estimate the trajectories of pixels across video frames. By capturing fine-grained motion, it enables dense understanding of scene dynamics and supports applications such as robotics (Vecerik et al., 2024), autonomous systems (Balasingam et al., 2024), and 4D scene generation (Wang et al., 2024; Lei et al., 2025). However, this task is inherently difficult due to large inter-frame displacements, motion blur, and occlusions in real-world videos, which hinder accurate trajectory estimation and reduce the robustness of existing approaches (Kim et al., 2025b).

To understand the limitations of point tracking, we first outline how conventional methods operate. Most approaches (Cho et al., 2024b; Karaev et al., 2024a) use a feature backbone to extract multi-scale features and predict an initial coarse trajectory (Doersch et al., 2023; Cho et al., 2024b). A local 4D cost volume is then built to encode correlations between the query and candidate points across space and time. A refiner updates the trajectory from this cost volume, which works well when the cost map attends correctly to the target point.

This framework, however, faces two key limitations. First, when the target point moves outside the receptive field due to large motion or occlusion, the local correlation becomes ambiguous (Xu et al., 2022; An et al., 2025). Second, because most point tracking models are trained primarily on synthetic data (Greff et al., 2022; Kim et al., 2025b; Balasingam et al., 2024), they often fail to generalize to real-world artifacts such as motion blur. While some methods incorporate real data through self-distillation (Karaev et al., 2024a; Doersch et al., 2024), these remain vulnerable to confirmation bias (Sohn et al., 2020).

To address both the structural limitation of local correlation maps and the domain gap from synthetic training data, we propose using video diffusion models based on Diffusion Transformers (DiTs) as robust feature backbones. Their internal 3D attention mechanism provides a global receptive field, directly handling large displacements and occlusions (Karaev et al., 2024a). In addition, their training on large-scale real-world videos equips them with stronger generalizability to visual artifacts. Given that video DiTs already demonstrate considerable zero-shot tracking ability (Nam et al., 2025), this motivates our central research question: *Can video DiTs address the fundamental challenges of point tracking?*

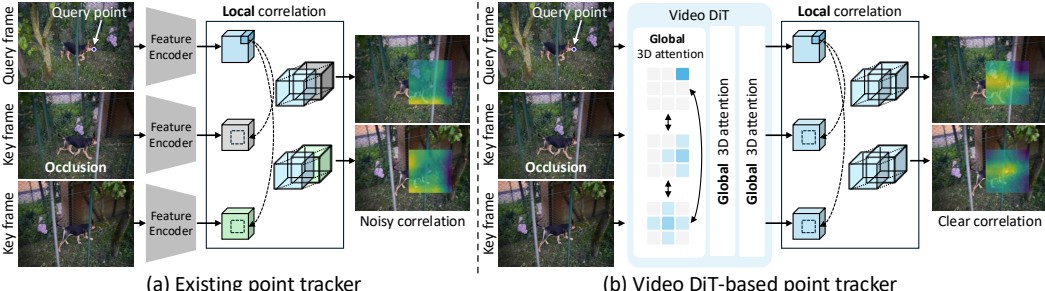

(a) Existing point tracker      (b) Video DiT-based point tracker

Figure 1: **Teaser.** In this work, we explore video diffusion model as a robust feature backbone for point tracking. (a) General point tracking backbone encodes each video frames independently and compute local correlation map, which struggles with challenging setting. (b) In contrast, video diffusion transformers (video DiTs) internally compute 3D global attention between entire video frames, which makes more temporally-consistent features and results in better correlation maps.

To investigate this, we compare video DiTs with the supervised backbone of CoTracker3. Despite the latter being explicitly trained for point tracking, video DiT features exhibit stronger temporal matching priors. We validate this by visualizing correlation maps under large displacements, where DiT features maintain consistent correspondences beyond the receptive field. We further evaluate trajectories under occlusion and synthetic blur, observing smaller performance drops than CoTracker3 (Karaev et al., 2024a) and DINOv2 (Oquab et al., 2023). These results confirm that video DITs offer superior temporal consistency and generalizability, making them strong backbones for point tracking.

Building on these findings, we extend DiT features into a supervised point tracking framework. To mitigate the coarse resolution of diffusion features, we introduce a shallow upsampler that fuses information across multiple layers and enhances spatial resolution. This design improves tracking accuracy compared to using DiT features alone. Moreover, under challenging conditions such as large motion, occlusion, and motion blur, our framework consistently outperforms CoTracker3, highlighting the effectiveness of combining diffusion-based features with supervised training.

In summary, our contributions are as follows:

- We present the first systematic study of video DiTs for point tracking, demonstrating their temporal consistency and robustness to large motion, occlusion, and motion blur.
- We extend DiT features into a supervised tracking framework with a lightweight upsampler that fuses multi-layer information and improves spatial resolution.
- We conduct extensive evaluations on challenging benchmarks, showing that our approach consistently outperforms CoTracker3 and other backbones under both standard and stress-test conditions.

## 2 RELATED WORK

**Point tracking.** Inspired by the classic Particle Video (Sand & Teller, 2008), PIPs (Harley et al., 2022) introduced deep-learning based point tracking by leveraging local correlation to iteratively refine estimates, a strategy widely adopted in optical flow tasks such as RAFT (Teed & Deng, 2020). TAPIR (Doersch et al., 2023) further advanced this approach by first computing global matches with TAP-Net (Doersch et al., 2022) and then refining them in a PIPs-style manner. Building on these foundations, subsequent works have expanded point tracking in various directions, including architectural modifications (Li et al., 2024b;a; Qu et al., 2024), multi-point interaction (Karaev et al., 2024b;a), adjustments to correlation receptive fields (Cho et al., 2024b), 3D extensions (Cho et al., 2025; Xiao et al., 2024), integration with optical flow (Cho et al., 2024a; Le Moing et al., 2024), and test-time optimization strategies (Tumanyan et al., 2024; Wang et al., 2023). Despite this progress, most methods remain trained exclusively on synthetic datasets (Greff et al., 2022), which limits robustness due to the domain gap with real-world scenarios. To address this, recent approaches such as BootsTAP (Doersch et al., 2024) and CoTracker3 (Karaev et al., 2024a) incorporate real-world videos via semi-supervised training, while other works (Kim et al., 2025b; Balasingam et al., 2024;

Jin et al., 2024) attempt to generate real-world tracking datasets, but these remain constrained by limited domain diversity or by the performance ceilings of current point trackers.

**Exploring feature backbone for point tracking.** Recent point tracking models have demonstrated substantial progress, with increasing emphasis on refinement modules to enhance predictive accuracy (Doersch et al., 2023; Cho et al., 2024b; Karaev et al., 2024b; Doersch et al., 2022). Nonetheless, most approaches continue to rely on fixed backbones such as ResNet or TSM-ResNet, leaving the potential of more expressive feature backbones underexplored. To address this, several works have investigated DINOv2 as a backbone, showing its strong effectiveness for point tracking (Kim et al., 2025a; Aydemir et al., 2025; Tumanyan et al., 2024). Furthermore, Aydemir et al. (2024) highlighted the broader promise of leveraging rich representations from diverse vision foundation models, with Stable Diffusion (Rombach et al., 2022) features even surpassing DINOv2 in tracking tasks. In line with this, DiffTrack (Nam et al., 2025) showed that video diffusion models, though not explicitly trained for tracking, contain layers well-suited for temporal correspondence and substantially outperform conventional backbones in zero-shot settings. Building on these insights, our work leverages video diffusion models to seamlessly integrate their knowledge into existing point tracking frameworks.

**Diffusion models for geometric tasks.** Building on the expressive representations learned through large-scale generative pre-training (Ho et al., 2020; Rombach et al., 2022), recent studies have shown that diffusion models capture strong geometric cues which can be adapted to various perception tasks, such as visual correspondence (Tang et al., 2023; Zhang et al., 2023; Meng et al., 2024; Gan et al., 2025; Nam et al., 2023), segmentation (Xu et al., 2023), and depth estimation (Ke et al., 2024). Pioneering works such as DIFT (Tang et al., 2023) and SD-DINO (Zhang et al., 2023) demonstrated that these models inherently encode semantic- and geometry-aware features, achieving competitive results on zero-shot correspondence tasks. Subsequent research has enhanced this capability through architectural modifications (Luo et al., 2023; Zhang et al., 2024; Xue et al., 2025; Liu et al., 2025), distillation strategies (Stracke et al., 2025), and prompt tuning (Li et al., 2024c), while largely preserving the original representation. Notably, this line of works hints that diffusion models can successfully generalize to perception tasks with only a handful of synthetic data (Ke et al., 2024), which can narrow the sim-to-real gap faced by conventional point tracking backbones. In this context, we extend further from DiffTrack (Nam et al., 2025), exploiting video diffusion features for point tracking exclusively on sparse, high-quality synthetic datasets.

## 3 METHOD

Conventional point tracking models struggle in challenging scnarios due to their structural limitation and dependece on synthetic training dataset. To mitigates these problem, we explore video DiTs as a strong candidate for point tracking feature backbone. To begin with, we briefly summarize recent correlation-based point tracking framework and attention mechanism of video DiTs in Section 3.1. Building on this, in Section 3.2, we analyze how 3D attention mechanism and a real-world prior of video DiTs can resolve the conventional limitations of a restricted receptive field and reliance on synthetic datasets, respctively. We extend our observations in Section 3.3, where we propose a bridging module training that effectively utilizes the powerful temporal features from a video DiTs for a point tracking task.

### 3.1 PRELIMINARIES

**Point tracking with local 4D correlation maps.** Given a video $X = \{I_i\}_{i=1}^{T}$, where each frame $I_i \in \mathbb{R}^{H \times W \times 3}$ with height $H$ and width $W$, point tracking is defined as estimating the trajectory of a query point $q = (i^q, x^q, y^q)$ specified in a reference frame $I_{i^q}$, where $x^q$ and $y^q$ denote spatial coordinates, and $i^q$ denotes a time index. The objective is to predict point positions $\{P_i = (x_i, y_i)\}_{i=1}^{T}$ together with visibility $\{V_i \in [0,1]\}_{i=1}^{T}$ and confidence $\{C_i \in [0,1]\}_{i=1}^{T}$. To this end, a feature encoder $\Phi(\cdot)$ first extracts dense feature maps $\Phi_i = \Phi(I_i) \in \mathbb{R}^{H/k \times W/k \times d}$ with ratio $k$ and dimension $d$, and a feature pyramid is constructed by average pooling feature $\Phi_i$ at $S$ different scales:

$$\Phi_i^s = \mathsf{Downsample}_s(\Phi_i) \in \mathbb{R}^{\frac{H}{k2^{s-1}} \times \frac{W}{k2^{s-1}} \times d}, \quad s = 1, \dots, S. \tag{1}$$

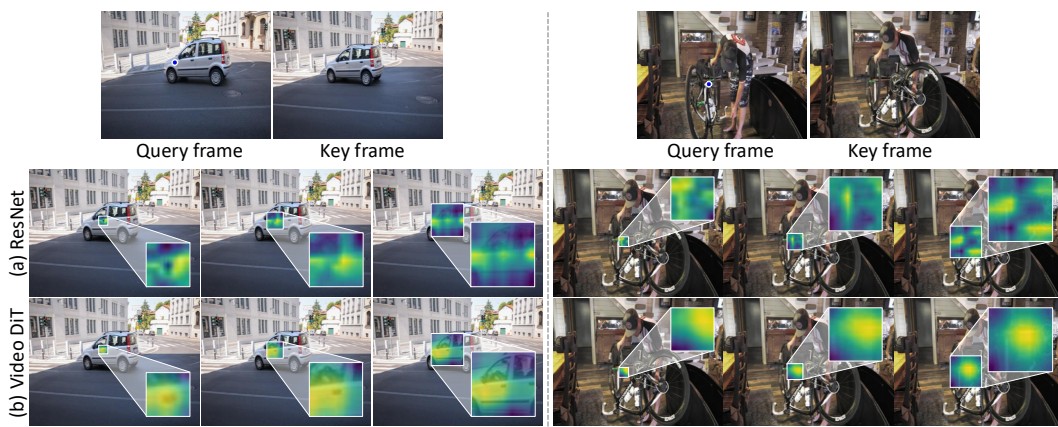

Figure 2: **Visualization of pyramid local costmaps.** For the query point on the left images, we compute feature similarity between the query point and key points from each image over multiple feature resolutions and visualize the cost map. (a) ResNet features often become noisy when the target point is on texturelss region or moves outside a local window. (b) Video DiT features, on the other hand, clearly highlight the direction of the true target point.

For each scale $s$, local features $\phi_i^s$ are sampled around the current estimate $P_i$ within a $\Delta$-sized neighborhood using bilinear interpolation:

$$\phi_i^s = \left[ \Phi_i^s \left( \tfrac{\mathbf{x}}{ks} + \delta, \ \tfrac{\mathbf{y}}{ks} + \delta \right) : \delta \in \mathbb{Z}, \ \|\delta\|_\infty \leq \Delta \right] \in \mathbb{R}^{d \times (2\Delta+1)^2}. \tag{2}$$

Given query features $\phi_{i^q}^s$ and target features $\phi_i^s$, a *local 4D correlation map* $\tilde{\mathcal{C}}_{i^q,i}^s$ is constructed by measuring pairwise similarity across spatial offsets and time:

$$\tilde{\mathcal{C}}_{i^q,i}^s = \phi_{i^q}^s (\phi_i^s)^\top \in \mathbb{R}^{(2\Delta+1)^4}. \tag{3}$$

This correlation volume encodes the likelihood of correspondence between the query and candidate points, and utilized to iteratively update the trajectory estimates $\{P_i, V_i, C_i\}$ in refinement module.

**Video Diffusion Transformers (DiTs).** Recent video diffusion models (Yang et al., 2024; Li et al., 2024d) demonstrate strong temporal consistency in video generation. DiffTrack (Nam et al., 2025) further shows that these models implicitly capture temporal correspondences: query–key similarities from selected layers can be interpreted as cost volumes that already yield competitive zero-shot tracking performance.

Formally, a video $X \in \mathbb{R}^{T \times H \times W \times 3}$ is encoded by a 3D VAE into latent representations $\mathbf{z}_\text{video} \in \mathbb{R}^{T' \times H' \times W' \times d}$, where $H' = H/c$, $W' = W/c$, and $T' = (T-1)/r + 1$ are determined by spatial compression ratio $c$, temporal compression ratio $r$, and latent dimension $d$. A text prompt is also encoded into $\mathbf{z}_\text{text}$ (Raffel et al., 2020), but is omitted here as we only focus on video correspondences.

A diffusion transformer (DiT) processes $\mathbf{z}_\text{video}$ through multiple layers of 3D attention. At each layer $l$ and head $h$, the latent in time frame $p$ is projected into query and key embeddings $Q_p^{l,h}$, $K_p^{l,h} \in \mathbb{R}^{H'W' \times d_h}$, with head dimension $d_h$. The attention mechanism then computes a matching cost $\mathcal{C}_{p,q}^{l,h}$ between latents at time frame $p$ and $q$:

$$\mathcal{C}_{p,q}^{l,h} = \mathsf{Softmax}\left( \frac{Q_p^{l,h}(K_q^{l,h})^\top}{\sqrt{d_h}} \right). \tag{4}$$

These attention-derived costs naturally encode temporal correspondences across frames, providing a strong prior for point tracking.

### 3.2 WHY VIDEO DIFFUSION TRANSFORMERS ENABLE ROBUST POINT TRACKING

We highlight two complementary advantages of video DiTs for point tracking: their 3D full-attention structure, which provides robustness to large motion and occlusion, and their real-world data prior,

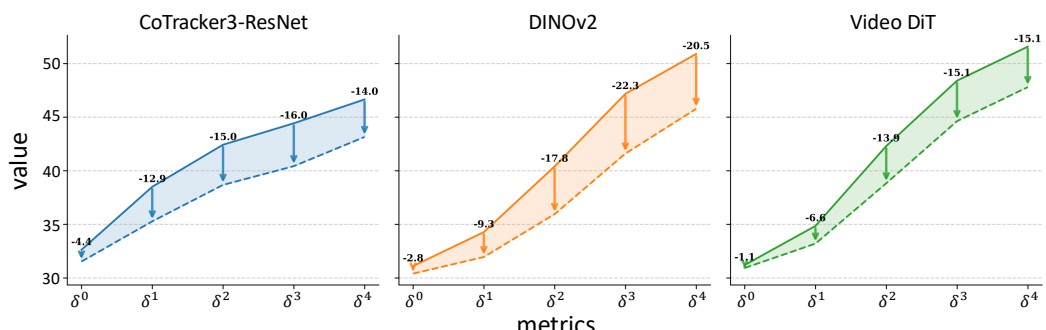

Figure 3: **Zero-shot performance for reappearing points on TAP-Vid-DAVIS dataset.** To evaluate robustness to occlusion, we anlayze various backbones exclusively on points that reappear after at least one occlusion. Video DiT and CoTracker3 proves the strongest robustness, wheareas DINOv2 is the most vulnerable. Notably, CoTracker3's robustness stems from its invisibility loss training.

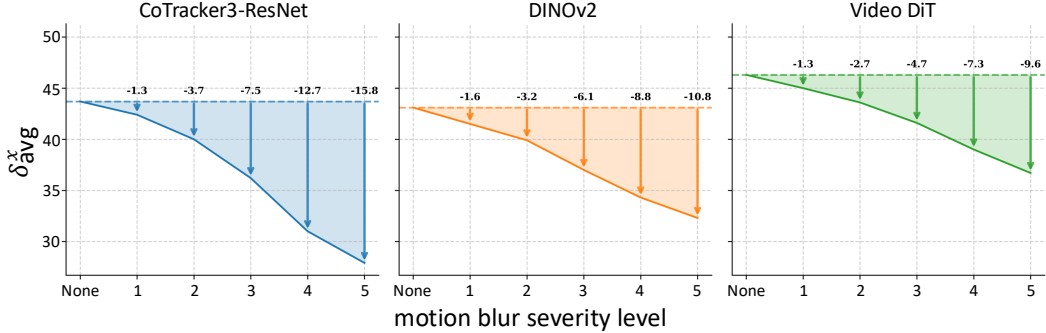

Figure 4: **Zero-shot performance on TAP-Vid-DAVIS dataset with motion blur augmentation.** To evaluate robustness to noise, we analyze various backbones on motion-blur augmented videos with varing severity. Video DiT demonstrates the highest robustness to noises.

which improves generalization to challenging visual conditions such as textureless regions and motion blur.

To illustrate these advantages, we first analyze local correlation maps that serve as the foundation of trajectory refinement. As shown in Fig. 7, ResNet features from CoTracker3 fail to construct reliable correlations in two scenarios: (1) textureless or repetitive regions, where the correlation becomes weak or ambiguous (Figure 7 (1)), and (2) large displacements, where the target point moves outside the local receptive field (Figure 7 (2)). In contrast, video DiT features produce sharper and more stable correlations, benefiting from both the robustness of the diffusion prior in data-scarce regions and the global temporal context captured by 3D attention.

We then validate these findings in downstream evaluations on occlusion and motion blur. For occlusion, we measure performance exclusively on points that reappear after being occluded. Fig. 3 shows that video DiT successfully re-identifies these points, outperforming both CoTracker3, which relies on an explicit invisibility loss, and DINOv2, which fails due to its frame-independent processing. This robustness arises naturally from the 3D attention mechanism (Yang et al., 2024), which integrates global temporal context across frames. For motion blur, we simulate varying blur levels to test robustness to noise. Fig. 4 demonstrates that video DiT suffers the smallest performance degradation, confirming that its large-scale real-world training endows it with strong generalization to visual artifacts (Nam et al., 2023).

Together, these results demonstrate that video diffusion models uniquely combine structural and data-driven advantages: 3D full attention that extends correlation beyond local receptive fields, and real-world priors that ensure resilience to noisy or ambiguous visual inputs. To further demonstrate the effectiveness of the video DiT, we compare its zero-shot tracking performance with the ResNet feature extractor from CoTracker3 (Karaev et al., 2024a). As shown in Table 1, the video DiT outperforms the CoTracker3 ResNet, despite the latter being explicitly trained with a ground-truth point tracking loss. This demonstrates the powerful inherent capabilities of video DiTs for robust point tracking.

Table 1: **Comparison between video DiT and CoTracker3-ResNet.** We evalute the zero-shot point tracking by comparing the video DiT (Nam et al., 2025) against the ResNet backbone of CoTracker3 (Karaev et al., 2024a), trained with full supervision on point tracking datasets. We further investigate the impact of feature resolution on tracking ability. All evaluation are conducted on TAP-Vid-Kinetics and TAP-Vid-DAVIS dataset (Doersch et al., 2022).

| Feature Backbone | Resolution | Kinetics | | | | | | DAVIS | | | | | |
|---|---|---|---|---|---|---|---|---|---|---|---|---|---|
| | | $< \delta^0$ | $< \delta^1$ | $< \delta^2$ | $< \delta^3$ | $< \delta^4$ | $< \delta^x_{\text{avg}}$ | $< \delta^0$ | $< \delta^1$ | $< \delta^2$ | $< \delta^3$ | $< \delta^4$ | $< \delta^x_{\text{avg}}$ |
| CoTracker3-ResNet | 96×128 | 9.8 | 30.0 | 48.0 | 57.0 | 64.7 | 41.9 | 10.5 | 34.0 | 49.7 | 57.7 | 66.6 | 43.7 |
| CoTracker3-ResNet | 48×64 | 14.6 | 30.1 | 45.9 | 56.3 | 65.2 | 42.4 | 10.9 | 27.6 | 45.3 | 56.2 | 67.7 | 41.5 |
| CoTracker3-ResNet | 24×32 | 1.0 | 3.7 | 14.8 | 36.4 | 49.3 | 21.0 | 0.7 | 2.6 | 13.0 | 29.3 | 45.3 | 18.2 |
| CoTracker3-ResNet | 12×16 | 0.1 | 0.5 | 2.1 | 8.3 | 29.4 | 8.1 | 0.1 | 0.3 | 1.2 | 6.0 | 24.8 | 6.5 |
| CoTracker3-ResNet | 30×45 | 2.4 | 9.4 | 25.4 | 39.4 | 50.3 | 25.4 | 1.7 | 6.5 | 21.0 | 34.2 | 48.2 | 22.3 |
| Video DiT (HunyuanVideo) | 30×45 | 5.9 | 22.0 | 49.1 | 70.4 | 80.3 | 45.5 | 4.4 | 18.2 | 44.8 | 70.1 | 82.8 | 44.1 |
| Video DiT (CogVideoX-2B) | 30×45 | 6.2 | 23.3 | 51.2 | 71.2 | 79.9 | 46.3 | 4.8 | 19.4 | 49.2 | 73.6 | 86.3 | 46.3 |
| Video DiT (CogVideoX-5B) | 30×45 | 6.8 | 25.9 | 55.4 | 74.9 | 82.7 | 49.2 | 5.2 | 20.5 | 50.7 | 73.9 | 84.3 | 46.9 |

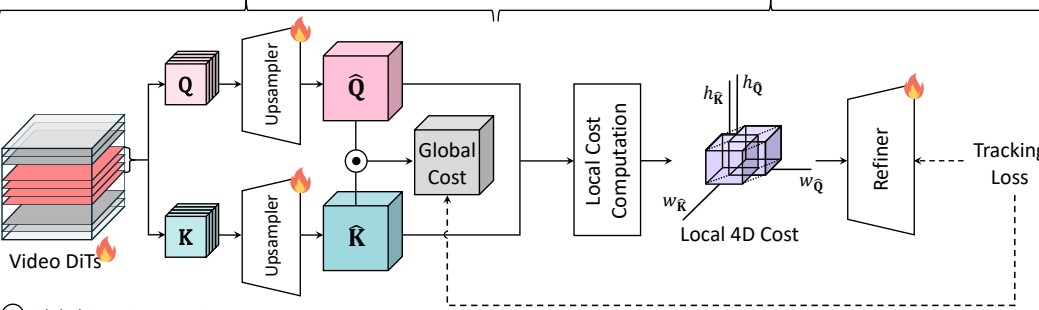

Figure 5: **Overall architecture for repurposing video DiTs for point tracking.** Our architecture consists of an upsampler and an iterative refiner. The upsampler increases feature resolution through training with a global correlation tracking loss. We then incorporate an iterative refiner that leverage local correlations from the upsampled feature to iteratively refine the predicted points.

## 3.3 REPURPOSING VIDEO DIFFUSION TRANSFORMERS FOR POINT TRACKING

Based on our analysis, we propose a novel point tracking network to fully utilize the video DiT features. Our method involves two main steps. We first create our feature backbone by adding an upsampler module to the video DiTs and training it. We then attach an iterative refiner to this backbone. The complete architecture is shown in Figure 5.

**Designing a bridging module.** In Table 1, Video DiT shows limited performance in fine-grained accuracy ($< \delta^0, < \delta^1$). Downsampled features of CoTracker3 also struggle to improve accuracy, consistent with prior findings that high-resolution backbones are necessary for precise matching (Edstedt et al., 2024; Cho et al., 2022; An et al., 2025).

To overcome this limitation, we propose an upsampler module that serves two purposes: (1) recovering high-resolution spatial detail from the compressed Video DiT features, and (2) fusing information across multiple layers that exhibit strong temporal consistency, as identified in prior work (Nam et al., 2025). Specifically, for each frame $i$, we extract queries and keys

$$Q_i = [Q_i^{l,h}]_{(l,h) \in \mathcal{S}}, \quad K_i = [K_i^{l,h}]_{(l,h) \in \mathcal{S}},$$

where $\mathcal{S}$ denotes the set of layer–head pairs with the highest temporal coherence. These descriptors are then increased resolution by upsampling module $\mathcal{U}(\cdot)$:

$$\hat{Q}_i = \mathcal{U}(Q_i), \quad \hat{K}_i = \mathcal{U}(K_i), \tag{5}$$

yielding high-resolution features $\hat{Q}_i, \hat{K}_i \in \mathbb{R}^{(1+f)H_u W_u \times d_h}$. We further build a feature pyramid $\{\hat{Q}_i^s, \hat{K}_i^s\}_{s=1}^S$ by average pooling, enabling multi-scale correlation matching.

**Point prediction using video DiT features.** Since the upsampler module is initialized from scratch, it requires training to learn how to effectively upsample and fuse high-resolution feature

Table 2: **Quantitative results on the TAP-Vid datasets (Doersch et al., 2022).** We evaluate performance improvements at each stage of the model. Incorporating the upsampler and then the iterative refiner leads to progressively more precise tracking.

| Upsamp. | Refiner | Kinetics | | | | | | DAVIS | | | | | |
|---------|---------|----------|----------|----------|----------|----------|---------------------|----------|----------|----------|----------|----------|---------------------|
| | | $< \delta^0$ | $< \delta^1$ | $< \delta^2$ | $< \delta^3$ | $< \delta^4$ | $< \delta^x_{\text{avg}}$ | $< \delta^0$ | $< \delta^1$ | $< \delta^2$ | $< \delta^3$ | $< \delta^4$ | $< \delta^x_{\text{avg}}$ |
| ✗ | ✗ | 6.2 | 23.3 | 51.2 | 71.2 | 79.9 | 46.3 | 4.8 | 19.4 | 29.2 | 73.6 | 86.3 | 46.3 |
| ✓ | ✗ | 14.3 | 28.7 | 42.2 | 53.2 | 63.2 | 40.3 | 16.0 | 38.5 | 64.6 | 82.0 | 91.3 | 58.5 |
| ✓ | ✓ | 15.2 | 29.3 | 42.1 | 52.6 | 62.0 | 40.2 | 18.9 | 42.0 | 66.9 | 84.1 | 91.9 | 60.7 |

maps. Given a query point $q = (i^q, x^q, y^q)$, we first extract the query feature $\mathbf{v}^s$ from the query maps $\hat{Q}^s_{i^q}$ of frame $i^q$ at position $(x^q, y^q)$ using bilinear interpolation for each scale $s$. For each frame $i$, we compute the correlation map $\mathcal{C}^s_{i^q,i}$ using cosine similarity between the query feature and the key feature maps:

$$\mathcal{C}^s_{i^q,i} = \frac{\mathbf{v}^{s\top}\hat{K}^s_i}{\|\mathbf{v}^s\|\|\hat{K}^s_i\|} \in \mathbb{R}^{\frac{H_u}{2^{s-1}} \times \frac{W_u}{2^{s-1}}}, \tag{6}$$

where $\|\cdot\|$ denotes L2 normalization. For each scale $s$, the point position $\hat{p}^s_i$ in frame $i$ is estimated with a soft-argmax operation over the correlation map:

$$\hat{p}^s_i = \sum_{(x,y)} \mathsf{Softmax}\big(\mathcal{C}^s_{i^q,i}(x,y)\big) \cdot (x,y), \tag{7}$$

where $(x, y)$ denotes pixel coordinates. We compute predictions at each scale and apply supervision independently to encourage consistent localization across resolutions. Following Kim et al. (2025a), we adopt the Huber loss (Huber, 1992) to supervise the predicted positions, which improves robustness to outliers. Loss is computed only on visible points.

**Adopting an iterative refiner.** Following conventional point tracking models that combine a feature backbone with an iterative refiner, we use video DiT with our upsampler module as the backbone to provide robust and precise initial matches. An iterative refiner is then applied to these initial predictions to enhance fine-grained accuracy. This two-stage design leverages the global robustness of video DiT features in challenging scenarios while exploiting a local refiner for high precision. For the refinement stage, we adopt CoTracker3's refiner (Karaev et al., 2024a), keeping the architecture and loss functions identical, except that we replace ResNet features $\phi^s_{i^q}, \phi^s_i$ with queries and keys $\hat{Q}^s_{i^q}, \hat{K}^s_i$ from video DiT when constructing local correlation volumes using Eq. 3.

## 4 Experiments

### 4.1 Implementation Details

**Training details.** We adopt CogVideoX-2B (Yang et al., 2024) as the feature backbone and extract features from the 13th, 17th, 18th, and 21st layers, which prior work identified as the most temporally coherent (Nam et al., 2025). A DPT head (Ranftl et al., 2021) is used to upsample and fuse these multi-layer query–key features. For efficient adaptation, we apply LoRA (Hu et al., 2022) with rank 128 to the video DiT. The LoRA–DPT backbone is trained for 10K steps, after which we attach the iterative refiner head from CoTracker3 (Karaev et al., 2024a). In this second stage, the LoRA parameters are frozen while the DPT head and refiner are trained jointly for 5K steps.

**Evaluation protocol.** We follow the TAP-Vid (Doersch et al., 2022) evaluation protocol on TAP-Vid-Kinetics, TAP-Vid-DAVIS datasets. For more details, please refer to Appendix A

### 4.2 Experimental results

**Ablation study.** Table 2 summarizes the performance of video DiT under different configurations. The zero-shot baseline already exhibits strong temporal matching ability without task-specific training. On DAVIS (Doersch et al., 2022), adding the upsampler yields consistent gains by recovering

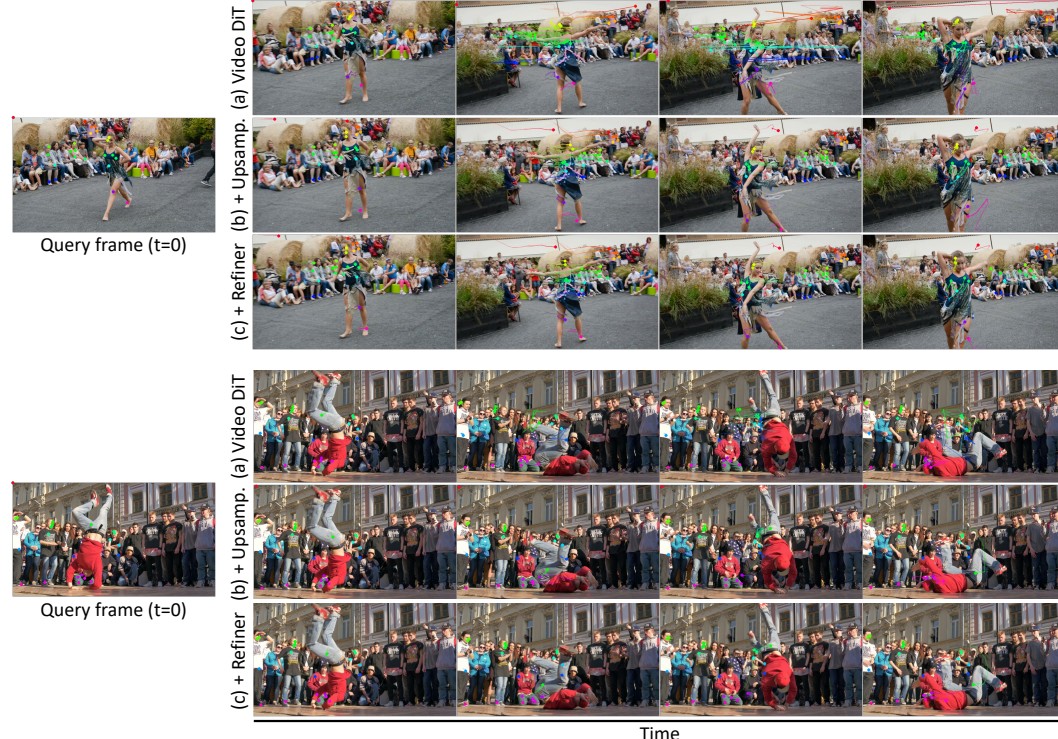

Figure 6: **Qualitative results of trajectory prediction.** (a) The pre-trained video DiTs produces a plausible correlation map that roughly indicates the motion direction of a queried point even when it lies outside the local window. (b–c) After fine-tuning, the model yields clearer and more accurate local correlation volumes, highlighting the effectiveness of our training approach.

Table 3: **Quantitative results of feature backbone on TAP-Vid dataset (Doersch et al., 2022).** We evaluate the global cost performance of our model with only the upsampler module attached, comparing it to other feature backbones.

| Backbone | Kinetics | | | | | | DAVIS | | | | | |
|---|---|---|---|---|---|---|---|---|---|---|---|---|
| | $< \delta^0$ | $< \delta^1$ | $< \delta^2$ | $< \delta^3$ | $< \delta^4$ | $< \delta^x_{\text{avg}}$ | $< \delta^0$ | $< \delta^1$ | $< \delta^2$ | $< \delta^3$ | $< \delta^4$ | $< \delta^x_{\text{avg}}$ |
| Tapir | 8.6 | 28.8 | 56.5 | 74.2 | 83.3 | 50.3 | 9.0 | 27.3 | 54.9 | 73.7 | 84.1 | 49.8 |
| TAP-Net | 7.8 | 28.1 | 55.2 | 71.4 | 80.2 | 48.6 | 7.3 | 23.1 | 46.7 | 66.6 | 79.2 | 44.6 |
| LocoTrack | 17.6 | 40.3 | 60.6 | 72.2 | 78.0 | 53.7 | 20.2 | 45.3 | 64.5 | 75.1 | 81.2 | 57.3 |
| CoTracker3 | 22.6 | 41.2 | 55.9 | 63.7 | 69.1 | 50.5 | 27.4 | 47.1 | 69.4 | 66.7 | 71.7 | 54.7 |
| Chrono | 26.0 | 48.4 | 68.2 | 79.8 | 85.3 | 61.6 | 26.1 | 52.6 | 74.5 | 84.9 | 90.0 | 65.6 |
| Video DiT | 6.2 | 23.3 | 51.2 | 71.2 | 79.9 | 46.3 | 4.8 | 19.4 | 49.2 | 73.6 | 86.3 | 46.3 |
| Video DiT + Upsamp. | 14.3 | 28.7 | 42.2 | 53.2 | 63.2 | 40.3 | 16.0 | 38.5 | 64.5 | 82.0 | 91.3 | 58.4 |

higher-resolution features, and incorporating the iterative refiner provides further improvements. Qualitative results in Figure 6 show that trajectories become increasingly accurate in challenging scenes as modules are added, while Figure 7 illustrates that local cost maps also become sharper and more reliable. These results highlight the complementary benefit of combining the global robustness of video DiT features with local refinement.

However, on Kinetics (Doersch et al., 2022), the model with supervision underperforms the zero-shot baseline. We attribute this gap to differences in video length: Kinetics contains much longer sequences than DAVIS, and our training setup with temporal interpolation in the upsampler was limited to shorter inputs. This mismatch suggests that future work should explore improved temporal modeling to better handle long video sequences.

**Comparison with feature backbones.** In Table 3, we compare our approach against existing feature backbones. Zero-shot video DiT already lags behind task-specific trackers such as Co-Tracker3 (Karaev et al., 2024a) and Chrono (Kim et al., 2025a) in fine-grained accuracy, reflect-

Table 4: **Quantitative results across different levels of noise severity on TAP-Vid-DAVIS dataset.** The results indicate that the superior real-world prior of video DiTs mitigates performance degradation, even under high levels of noise.

| Methods | Original | | | Severity 1 | | | Severity 3 | | | Severity 5 | | |
|---|---|---|---|---|---|---|---|---|---|---|---|---|
| | AJ↑ | $< \delta^x_{avg}$ ↑ | OA↑ | AJ↑ | $< \delta^x_{avg}$ ↑ | OA↑ | AJ↑ | $< \delta^x_{avg}$ ↑ | OA↑ | AJ↑ | $< \delta^x_{avg}$ ↑ | OA↑ |
| CoTracker3 | 64.4 | 76.9 | 91.2 | 62.8 | 75.2 | 90.4 | 54.4 | 67.5 | 89.0 | 47.1 | 60.4 | 86.2 |
| | (-) | (-) | (-) | (-1.6) | (-1.7) | (-0.8) | (-10.0) | (-9.4) | (-2.2) | (-17.3) | (-16.5) | (-5.0) |
| Ours | 42.5 | 60.9 | 72.2 | 42.2 | 59.8 | 72.5 | 39.5 | 55.6 | 72.4 | 36.1 | 51.8 | 71.5 |
| | (-) | (-) | (-) | (-0.3) | (-1.1) | (+0.3) | (-3.0) | (-5.3) | (+0.2) | (-6.4) | (-9.1) | (-0.7) |

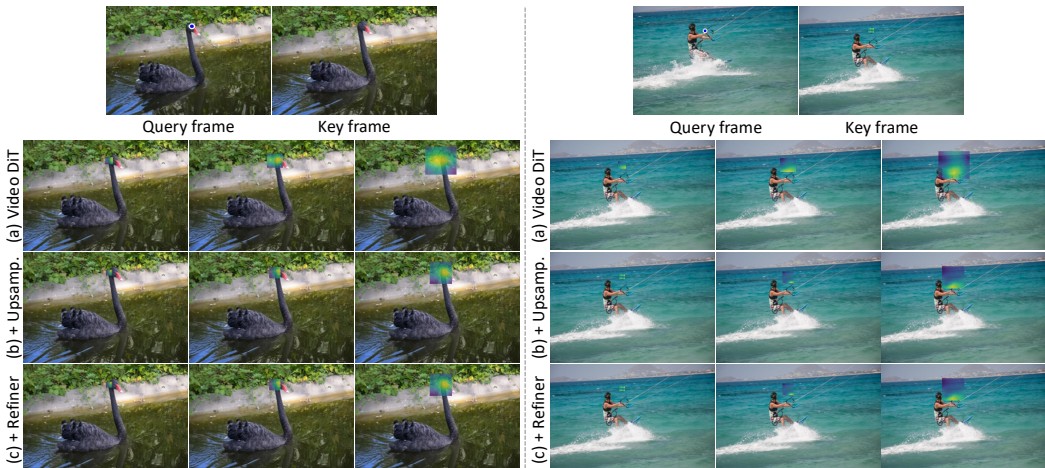

Figure 7: **Qualitative results of local correlation.** (a) The pre-trained video DiT produces a plausible correlation map that roughly indicates the motion direction of a queried point even when it lies outside the local window. (b–c) After fine-tuning, the model yields clearer and more accurate local correlation volumes, highlighting the effectiveness of our training approach.

ing its coarse resolution. However, equipping video DiT with the upsampler substantially improves performance, narrowing the gap with supervised backbones. On DAVIS, video DiT with upsampling module achieves $58.4\%$ under $\delta^x_{avg}$, surpassing both TAP-Net (Doersch et al., 2022) and Loco-Track (Cho et al., 2024b). These results demonstrate that video DiT features, when paired with an upsampling module, provide a strong alternative to supervised backbones for point tracking.

**Analysis on robustness in motion blur.** We evaluate robustness to motion blur by measuring performance across varying blur severities, with results presented in Table 4. While the performance of CoTracker3 degrades significantly as blur intensity increases, our model remains much more stable. These findings highlights that the self-distillation method used by CoTracker3 is not sufficient to inject robustness to real-world artifacts. On the other hand, the strong real-world priors inherent in the video DiT provide superior resilience to this kind of noise. This robustness creates a more reliable correlation map, which prevents significant error propagation in the iterative refiner.

## 5 CONCLUSION

This work establishes video DiTs as a strong backbone for point tracking. Through systematic analysis, we demonstrated that their 3D full-attention structure mitigates failures from local correlation limits, while their large-scale real-world training provides resilience to visual artifacts such as motion blur. Building on these insights, we proposed a bridging module that upsamples and fuses DiT features before applying an iterative refiner, yielding consistently stronger performance across benchmarks. While our current training pipeline underperforms on very long sequences due to temporal compression, this highlights a promising direction for future work on temporal scaling. Overall, our findings suggest that video DiTs offer a scalable and generalizable foundation for robust point tracking, opening the door to further integration of generative video models into geometric perception tasks.

## REPRODUCIBILITY STATEMENT

We detail the training configurations in Section 4.1 and Appendix A. We will also release our code and model checkpoints to ensure reproducibility.

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

## A FURTHER IMPLEMENTATION DETAILS

**Additional training details.** Both training stages use AdamW (Loshchilov & Hutter, 2017) with a learning rate of $5 \times 10^{-4}$, weight decay $5 \times 10^{-4}$, and a cosine schedule with 500 warm-up steps. Training is conducted on the TAP-Vid-Kubric dataset (Greff et al., 2022) for a total of 15K iterations using 4 NVIDIA A6000 GPUs. We sample videos of length $T \in 30, \ldots, 60$ and uniformly choose 512 query points per video. The batch size is set to 1 with gradient accumulation of 4, yielding an effective batch size of 16. All frames are resized to $480 \times 720$ to match the optimal input resolution of CogVideoX-2B.

**Evaluation protocol.** We follow the TAP-Vid (Doersch et al., 2022) evaluation protocol on TAP-Vid-Kinetics, TAP-Vid-DAVIS datasets. TAP-Vid-Kinetics comprises 1,144 YouTube videos from the Kinetics-700-2020 (Carreira & Zisserman, 2017) validation set with an average of 26 tracks per video. TAP-Vid-DAVIS contains 30 videos from the DAVIS 2017 (Pont-Tuset et al., 2017) with an average of 22 tracks per video. As evaluation metrics for the feature backbone, we report position accuracy at five threshold levels (Doersch et al., 2022; Kim et al., 2025a) ($\delta^0$, $\delta^1$, $\delta^2$, $\delta^3$, $\delta^4$), corresponding to pixel distances of 1, 2, 4, 8, and 16, repectively. We also report the average accruacy across all threshold ($\delta_{\text{avg}}^x$). For the noise setting, we evaluate the robustness by systemically injecting motion blur with different level of severity (Hendrycks & Dietterich, 2019). In this setting, we use Average Jaccard (AJ), position accuracy ($\delta_{\text{avg}}^x$), and occlusion accuracy (OA) as metric.

## B USE OF LARGE LANGUAGE MODELS

In accordance with the ICLR 2026 submission policy, we disclose that Large Language Models were used to correct grammar and polishing of the writing.

