# OpenReview forum: "Video Diffusion Model for Point Tracking"
_ICLR.cc/2026/Conference — ICLR 2026 Conference Withdrawn Submission_

### Official Review · Reviewer_Mruo · 2025-10-26

**Soundness:** 2
**Presentation:** 3
**Contribution:** 1
**Rating:** 2
**Confidence:** 5

**Summary:**

The paper explores video diffusion transformers (DiTs) as a feature backbone for point tracking. It first analyzes DiT features, showing more stable correlation maps under large motion, occlusion, and blur. Building on this, the authors introduce a lightweight upsampler to recover spatial detail from DiT tokens and plug it into an iterative refinement tracking pipeline. Experiments are conducted on TAP-Vid (Kinetics and DAVIS).

**Strengths:**

* **Strong backbone choice**. Leveraging a video DiT backbone—pretrained at scale on real videos—provides a compelling feature for point tracking.

* **Clear feature analysis**. The paper provides a detailed comparison against ResNet/DINO backbones, showing that DiT features yield more stable, higher-quality correlation maps under large motion, occlusion, and blur, mitigating the limits of local cost-volume matching.

**Weaknesses:**

* **Limited technical novelty**. The paper’s main contribution is to replace a ResNet backbone with a DiT backbone plus an upsampler, rather than leveraging diffusion as a generative pipeline to for point tracking. This reads more like an engineering effort than a substantive research contribution.

* The paper states: “First, when the target point moves outside the receptive field due to large motion or occlusion, the local correlation becomes ambiguous.” I do not believe this can be fixed simply by using a stronger backbone (DiT). If the target truly moves beyond the correlation’s receptive field, the tracker lacks the necessary evidence to infer motion—regardless of feature quality. Addressing this requires expanding the receptive field (e.g., multi-scale/global correlation) or changing the architecture to incorporate global attention / long-range matching, not merely swapping in DiT features. Therefore, I don't agree with the claim that “video DiTs can address this fundamental challenge of point tracking”.

* **Limited evaluation**. Despite claiming strong generalization from video diffusion, the experiments are confined to two TAP-Vid subsets, and the method performs better on one, and worse on other, undermining the generalization claim.

* Reported results are far below prior work (e.g., CoTracker3: $< \delta^{x}_{\mathrm{avg}}$ on Kinetics/DAVIS of 67.8/76.9), yet Table 3 shows much lower values here; is there any different evaluation protocol for this.

* **Unclear takeaways**. On Kinetics, fine-tuning + upsampler hurts performance, while on DAVIS it helps. These contradictory outcomes make it hard to draw conclusions—should practitioners use a DiT backbone for tracking, and should it be fine-tuned?

* **Runtime trade-offs not justified**. Compared with lightweight ResNet-based trackers (e.g., CoTracker3, LocoTrack), the DiT backbone is slower. Given the mixed accuracy results, it’s unclear that the added compute is a good trade-off—especially for real-world deployments where tracking remains a latency-sensitive, low-level vision component.

**Questions:**

* Can the authors further evaluate both the zero-shot model and the fine-tuned + upsampler variant on additional test sets such as DynamicReplica, RoboTAP, and RGB-Stacking [1][2]?

* Can the authors report the runtime and peak GPU memory of the proposed method? How many points can your model support in a single forward pass?

* My take is that the pretrained DiT backbone shows impressive correspondence ability (key for tracking), though it still lags well-trained tracking methods. A more suitable adaptation seems necessary. It appears the current bottleneck is the fine-tuning setup and the upsampler design. Could the authors try different upsamplers and fine-tuning strategies to close the gap?

**Conclusion**

While the paper is well-motivated and the DiT backbone is a reasonable choice for robust features, the work reads largely as a backbone swap, and the evidence does not support the stated generalization benefits. Evaluation is limited and findings are inconclusive about fine-tuning and the upsampler, and the runtime trade-off is not reported. Based on this, I recommend reject.

[1] Karaev, Nikita et al. “CoTracker3: Simpler and Better Point Tracking by Pseudo-Labelling Real Videos.” ArXiv abs/2410.11831 (2024)

[2] Harley, Adam W. et al. “AllTracker: Efficient Dense Point Tracking at High Resolution.” ArXiv abs/2506.07310 (2025)

---

### Official Review · Reviewer_2L3f · 2025-10-26

**Soundness:** 1
**Presentation:** 2
**Contribution:** 1
**Rating:** 2
**Confidence:** 4

**Summary:**

This paper conducts a further exploration of DiffTrack, offering a more in-depth comparison between video diffusion transformer (DiT) models and traditional vision backbones. However, overall, the work appears to be an incremental extension of DiffTrack rather than a substantially new contribution. Neither the insight nor the technical innovation is sufficiently novel to meet the ICLR standards for contribution and originality.

**Strengths:**

The paper is well-written and easy to follow. The contributions are clearly claimed, and the corresponding supporting experiments are provided.

**Weaknesses:**

The overall insight of the paper is not novel.

1. The idea of analyzing temporal correspondences across frames using diffusion transformers (DiTs) from a point tracking perspective has already been explored in DiffTrack.

2. In terms of methodology, the contribution appears quite limited. The proposed approach essentially builds on DiffTrack by adding an upsampler and concatenating it with the Refiner module from CoTracker.

3. The main technical contribution is the upsampler, which is simply designed to rescale the multi-scale features proposed in DiffTrack to a uniform resolution, followed by feature pooling. However, it seems that there is no clear or formal definition of the upsampler module in the paper.

4. While Table 3 shows improvement on DAVIS, that dataset contains only 30 videos. On the more challenging and large-scale Kinetics dataset, the proposed method even leads to significant performance degradation. These results fail to convincingly demonstrate the effectiveness of the upsampler.

**Questions:**

Please refer to the weakness.

---

### Official Review · Reviewer_3P9a · 2025-10-28

**Soundness:** 2
**Presentation:** 2
**Contribution:** 1
**Rating:** 2
**Confidence:** 5

**Summary:**

This paper explores the use of video diffusion transformers (Video DiTs) as feature backbones for point tracking, aiming to leverage their global 3D attention and large-scale real-world pretraining to improve robustness under large motion, occlusion, and motion blur. The authors propose a simple two-stage adaptation: (i) an upsampler module to recover spatial detail and fuse multi-layer DiT features; (ii) an iterative refiner adopted from common point trackers for high-precision trajectory estimation. Experiments on TAP-Vid DAVIS and Kinetics datasets are reported.

**Strengths:**

* Diffusion models are indeed a natural candidate for geometric perception tasks, and this work provides an initial empirical exploration of how their temporal reasoning could benefit pixel-level correspondence.
* Conceptually simple and clear idea. The paper integrates video DiTs into a point tracking framework with minimal modifications and clear motivation.
* Well-written and easy to follow, with coherent structure and extensive visualizations (cost volumes, qualitative examples).

**Weaknesses:**

* While the paper claims that Video DiTs inherently capture temporal consistency, results (Table 3) show that they underperform most image-based backbones, even when equipped with the upsampler and refiner. Given that image-based backbones model temporal relations after feature extraction, Video DiTs should have an advantage in these zero-shot settings. Yet, their performance remains lower, contradicting the claimed benefit of 3D attention in Section 3.2.
* The proposed upsampler leads to inconsistent gains across datasets: improvements on DAVIS but a drop on Kinetics. This suggests that the design may not generalize well and the gains are dataset-specific. The paper should analyze why the upsampler struggles in long or diverse sequences.
* Evaluation is limited to two datasets (DAVIS, Kinetics). To establish the generality of diffusion-based backbones, additional domains, such as RoboTAP (robotics), should be tested. These would better reflect the claim of robustness to real-world artifacts.
* The scale gap between Video DiTs (billions of parameters) and ResNet-based CoTracker3 backbones (tens of millions) raises fairness concerns. The current results may conflate improvements from model capacity rather than architectural superiority. The authors should either (a) compare against equally large vision backbones (eg DINOv3 7B, SwinV2 3B), or (b) use smaller DiTs to isolate the temporal modeling effect.
* Video DiTs are known for high memory and computational cost. Section A mentions 4 A6000 GPUs for 15 K iterations for training, but memory usage and runtime are not quantified for inference.

### Overall
The paper explores a promising direction, integrating video diffusion transformers into point tracking, but lacks strong empirical evidence and insight on why and how this integration improves performance. Most observed gains are modest or dataset-specific, and core claims about temporal reasoning and real-world robustness are not convincingly demonstrated. If extended with deeper analysis of DiT adaptation, this line of work could meaningfully contribute to understanding how generative video priors support geometric perception.

**Questions:**

See weaknesses.

---

### Official Review · Reviewer_fAdW · 2025-11-01

**Soundness:** 3
**Presentation:** 3
**Contribution:** 1
**Rating:** 2
**Confidence:** 4

**Summary:**

The paper proposes using features from video diffusion transformers as a backbone for point tracking. On TAP-Vid dataset, the method shows strong robustness and competitive accuracy. Zero-shot DiT features already outperform a supervised ResNet backbone under occlusion and motion blur condition, and the upsampler + refiner improve fine localization.

**Strengths:**

1. This paper presents a study on video DiTs for point tracking and demonstrates that it outperforms ResNet under conditions involving large motion, occlusion, and motion blur.
2. The authors conducted extensive evaluations showing that their backbone surpasses CoTracker3 and other competing backbones.
3. The paper is well-written and easy to follow.

**Weaknesses:**

1. This paper does not introduce a new method. It only replaces the standard backbone with a Video DiT.
2. Table 4 claims that Video DiTs mitigate performance degradation. However, CoTracker3 still achieves much better performance, even under greater degradation.
3. The Video DiT model is computationally heavy and requires much more GPU memory.

**Questions:**

1. The refiner is expected to significantly improve performance in other models; however, according to Table 2, it only provides a slight improvement here. Could you explain why?
2. Could you provide results for the other complete model, not just the backbone?
3. Could you clarify what you mean by zero-shot point tracking?

---

### Note · Authors · 2025-11-12

I have read and agree with the venue's withdrawal policy on behalf of myself and my co-authors.